# Towards Playing Full MOBA Games with Deep Reinforcement Learning

**Deheng Ye**[1]   **Guibin Chen**[1]   **Wen Zhang**[1]   **Sheng Chen**[1]   **Bo Yuan**[1]   **Bo Liu**[1]   **Jia Chen**[1]
**Zhao Liu**[1]   **Fuhao Qiu**[1]   **Hongsheng Yu**[1]   **Yinyuting Yin**[1]   **Bei Shi**[1]   **Liang Wang**[1]
**Tengfei Shi**[1]   **Qiang Fu**[1]   **Wei Yang**[1]   **Lanxiao Huang**[2]   **Wei Liu**[1]

[1] Tencent AI Lab, Shenzhen, China
[2] Tencent TiMi L1 Studio, Chengdu, China

{dericye,beanchen,zivenwzhang,victchen,jerryyuan,leobliu,jaylahchen,ricardoliu,
frankfhqiu,yannickyu,mailyyin,beishi,enginewang,francisshi,leonfu,
willyang,jackiehuang}@tencent.com; wl2223@columbia.edu

## Abstract

MOBA games, e.g., Honor of Kings, League of Legends, and Dota 2, pose grand challenges to AI systems such as multi-agent, enormous state-action space, complex action control, etc. Developing AI for playing MOBA games has raised much attention accordingly. However, existing work falls short in handling the raw game complexity caused by the explosion of agent combinations, i.e., lineups, when expanding the hero pool in case that OpenAI's Dota AI limits the play to a pool of only 17 heroes. As a result, full MOBA games without restrictions are far from being mastered by any existing AI system. In this paper, we propose a MOBA AI learning paradigm that methodologically enables playing full MOBA games with deep reinforcement learning. Specifically, we develop a combination of novel and existing learning techniques, including curriculum self-play learning, policy distillation, off-policy adaption, multi-head value estimation, and Monte-Carlo tree-search, in training and playing a large pool of heroes, meanwhile addressing the scalability issue skillfully. Tested on Honor of Kings, a popular MOBA game, we show how to build superhuman AI agents that can defeat top esports players. The superiority of our AI is demonstrated by the first large-scale performance test of MOBA AI agent in the literature.

## 1 Introduction

Artificial Intelligence for games, a.k.a. Game AI, has been actively studied for decades. We have witnessed the success of AI agents in many game types, including board games like Go [30], Atari series [21], first-person shooting (FPS) games like Capture the Flag [15], video games like Super Smash Bros [6], card games like Poker [3], etc. Nowadays, sophisticated strategy video games attract attention as they capture the nature of the real world [2], e.g., in 2019, *AlphaStar* achieved the grandmaster level in playing the general real-time strategy (RTS) game - StarCraft 2 [33].

As a sub-genre of RTS games, Multi-player Online Battle Arena (MOBA) has also attracted much attention recently [38, 36, 2]. Due to its playing mechanics which involve multi-agent competition and cooperation, imperfect information, complex action control, and enormous state-action space, MOBA is considered as a preferable testbed for AI research [29, 25]. Typical MOBA games include *Honor of Kings*, *Dota*, and *League of Legends*. In terms of complexity, a MOBA game, such as *Honor of Kings*, even with significant discretization, could have a state and action space of magnitude $10^{20000}$ [36], while that of a conventional Game AI testbed, such as Go, is at most $10^{360}$ [30]. MOBA games are further complicated by the real-time strategies of multiple heroes (each hero is uniquely

designed to have diverse playing mechanics), particularly in the 5 versus 5 (5v5) mode where two teams (each with 5 heroes selected from the hero pool) compete against each other [1].

In spite of its suitability for AI research, mastering the playing of MOBA remains to be a grand challenge for current AI systems. State-of-the-art work for MOBA 5v5 game is *OpenAI Five* for playing *Dota 2* [2]. It trains with self-play reinforcement learning (RL). However, *OpenAI Five* plays with one major limitation [2], i.e., only 17 heroes were supported, despite the fact that the hero-varying and team-varying playing mechanism is the soul of MOBA [38, 29].

As the most fundamental step towards playing full MOBA games, scaling up the hero pool is challenging for self-play reinforcement learning, because the number of agent combinations, i.e., lineups, grows polynomially with the hero pool size. The agent combinations are 4,900,896 ($C_{17}^{10} \times C_{10}^5$) for 17 heroes, while exploding to 213,610,453,056 ($C_{40}^{10} \times C_{10}^5$) for 40 heroes. Considering the fact that each MOBA hero is unique and has a learning curve even for experienced human players, existing methods by randomly presenting these disordered hero combinations to a learning system can lead to "learning collapse" [1], which has been observed from both *OpenAI Five* [2] and our experiments. For instance, OpenAI attempted to expand the hero pool up to 25 heroes, resulting in unacceptably slow training and degraded AI performance, even with thousands of GPUs (see Section "More heroes" in [24] for more details). Therefore, we need MOBA AI learning methods that deal with scalability-related issues caused by expanding the hero pool.

In this paper, we propose a learning paradigm for supporting full MOBA game-playing with deep reinforcement learning. Under the actor-learner pattern [12], we first build a distributed RL infrastructure that generates training data in an off-policy manner. We then develop a unified actor-critic [20] network architecture to capture the playing mechanics and actions of different heroes. To deal with policy deviations caused by the diversity of game episodes, we apply off-policy adaption, following that of [38]. To manage the uncertain value of state-action in game, we introduce multi-head value estimation into MOBA by grouping reward items. Inspired by the idea of curriculum learning [1] for neural network, we design a curriculum for the multi-agent training in MOBA, in which we "start small" and gradually increase the difficulty of learning. Particularly, we start with fixed lineups to obtain teacher models, from which we distill policies [26], and finally we perform merged training. We leverage student-driven policy distillation [9] to transfer the knowledge from easy tasks to difficult ones. Lastly, an emerging problem with expanding the hero pool is drafting, a.k.a. hero selection, at the beginning of a MOBA game. The Minimax algorithm [18] for drafting used in existing work with a small-sized hero pool [2] is no longer computationally feasible. To handle this, we develop an efficient and effective drafting agent based on Monte-Carlo tree search (MCTS) [7].

Note that there still lacks a large-scale performance test of Game AI in the literature, due to the expensive nature of evaluating AI agents in real games, particularly for sophisticated video games. For example, *AlphaStar Final* [33] and *OpenAI Five* [2] were tested: 1) against professionals for 11 matches and 8 matches, respectively; 2) against the public for 90 matches and for 7,257 matches, respectively (all levels of players can participate without an entry condition). To provide more statistically significant evaluations, we conduct a large-scale MOBA AI test. Specifically, we test using *Honor of Kings*, a popular and typical MOBA game, which has been widely used as a testbed for recent AI advances [36, 38, 37]. AI achieved a 95.2% win-rate over 42 matches against professionals, and a 97.7% win-rate against players of High King level [3] over 642,047 matches.

To sum up, our contributions are:

- We propose a novel MOBA AI learning paradigm towards playing full MOBA games with deep reinforcement learning.

- We conduct the first large-scale performance test of MOBA AI agents. Extensive experiments show that our AI can defeat top esports players.

## 2 Related Work

Our work belongs to system-level AI development for strategy video game playing, so we mainly discuss representative works along this line, covering RTS and MOBA games.

**General RTS games** StarCraft has been used as the testbed for Game AI research in RTS for many years. Methods adopted by existing studies include rule-based, supervised learning, reinforcement learning, and their combinations [23, 34]. For rule-based methods, a representative is *SAIDA*, the champion of StarCraft AI Competition 2018 (see `https://github.com/TeamSAIDA/SAIDA`). For learning-based methods, recently, *AlphaStar* combined supervised learning and multi-agent reinforcement learning and achieved the grandmaster level in playing StarCraft 2 [33]. Our value estimation (Section 3.2) shares similarity to *AlphaStar*'s by using invisible opponent's information.

**MOBA games** Recently, a macro strategy model, named *Tencent HMS*, was proposed for MOBA Game AI [36]. Specifically, HMS is a functional component for guiding where to go on the map during the game, without considering the action execution of agents, i.e., micro control or micro-management in esports, and is thus not a complete AI solution. The most relevant works are *Tencent Solo* [38] and *OpenAI Five* [2]. Ye et al. [38] performed a thorough and systematic study on the playing mechanics of different MOBA heroes. They developed a RL system that masters the micro control of agents in MOBA combats. However, only 1v1 solo games were studied without the much more sophisticated multi-agent 5v5 games. On the other hand, the similarities between this work and Ye et al. [38] include: the modeling of action heads (the value heads are different) and off-policy correction (adaption). In 2019, OpenAI introduced an AI for playing 5v5 games in Dota 2, called *OpenAI Five*, with the ability to defeat professional human players [2]. *OpenAI Five* is based on deep reinforcement learning via self-play. It trains using Proximal Policy Optimization (PPO) [28]. The major difference between our work and *OpenAI Five* is that the goal of this paper is to develop AI programs towards playing full MOBA games. Hence, methodologically, we introduce a set of techniques of off-policy adaption, curriculum self-play learning, value estimation, and tree-search that addresses the scalability issue in training and playing a large pool of heroes. On the other hand, the similarities between this work and *OpenAI Five* include: the design of action space for modeling MOBA hero's actions, the use of recurrent neural network like LSTM for handling partial observability, and the use of one model with shared weights to control all heroes.

## 3 Learning System

To address the complexity of MOBA game-playing, we use a combination of novel and existing learning techniques for neural network architecture, distributed system, reinforcement learning, multi-agent training, curriculum learning, and Monte-Carlo tree search. Although we use *Honor of Kings* as a case study, these proposed techniques are also applicable to other MOBA games, as the playing mechanics across MOBA games are similar.

### 3.1 Architecture

MOBA can be considered as a multi-agent Markov game with partial observations. Central to our AI is a policy $\pi_\theta(a_t|s_t)$ represented by a deep neural network with parameters $\theta$. It receives previous observations and actions $s_t = o_{1:t}, a_{1:t-1}$ from the game as inputs, and selects actions $a_t$ as outputs. Internally, observations $o_t$ are encoded via convolutions and fully-connected layers, then combined as vector representations, processed by a deep sequential network, and finally mapped to a probability distribution over actions. The overall architecture is shown in Fig. 1.

The architecture consists of general-purpose network components that model the raw complexity of MOBA games. To provide informative observations to agents, we develop multi-modal features, consisting of a comprehensive list of both scalar and spatial features. Scalar features are made up of observable units' attributes, in-game statistics and invisible opponent information, e.g., health point (hp), skill cool down, gold, level, etc. Spatial features consist of convolution channels extracted from hero's local-view map. To handle partial observability, we resort to LSTM [14] to maintain memories between steps. To help target selection, we use target attention [38, 2], which treats the encodings after LSTM as query, and the stack of game unit encodings as attention keys. To eliminate unnecessary RL explorations, we design action mask, similar to [38]. To manage the combinatorial

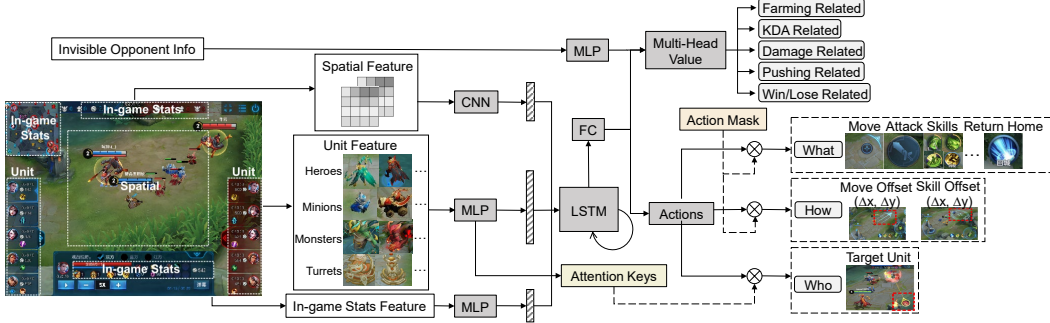

Figure 1: Our neural network architecture.

action space of MOBA, we develop hierarchical action heads and discretize each head. Specifically, AI predicts the output actions hierarchically: 1) what action to take, e.g., move, attack, skill releasing, etc; 2) who to target, e.g., a turret or an enemy hero or others; 3) how to act, e.g., a discretized direction to move.

## 3.2 Reinforcement Learning

We use the actor-critic paradigm [20], which trains a value function $V_\theta(s_t)$ with a policy $\pi_\theta(a_t|s_t)$. And we use off-policy training, i.e., updates are applied asynchronously on replayed experiences.

MOBA game with a large hero pool poses several challenges when viewed as a reinforcement learning problem: off-policy learning can be unstable due to long-term horizons, combinatorial action space and correlated actions; a hero and its surroundings are evolving and ever-changing during the game, making it difficult to design reward and estimate the value of states and actions.

**Policy updates**. We assume independence between action heads, so as to simplify the correlations between action heads, e.g., the direction of a skill ("How") is conditioned on the skill type ("What"), which is similar to [38, 33]. In our large-scale distributed environment, the trajectories are sampled from various sources of policies, which can differ considerably from the current policy $\pi_\theta$. To avoid training instability, we use Dual-clip PPO [38], which is an off-policy optimized version of the PPO algorithm [28]. Considering that when $\pi_\theta(a_t^{(i)}|s_t) \gg \pi_{\theta_{old}}(a_t^{(i)}|s_t)$ and the advantage $\hat{A}_t < 0$, ratio $r_t(\theta) = \frac{\pi_\theta(a_t|s_t)}{\pi_{\theta_{old}}(a_t|s_t)}$ will introduce a big and unbounded variance since $r_t(\theta)\hat{A}_t \ll 0$. To handle this, when $\hat{A}_t < 0$, Dual-clip PPO introduces one more clipping hyperparameter $c$ in the objective:

$$\mathcal{L}^{policy}(\theta) = \hat{\mathbb{E}}_t\Big[\max\Big(\min\Big(r_t(\theta)\hat{A}_t, \text{clip}\big(r_t(\theta), 1-\epsilon, 1+\epsilon\big)\hat{A}_t\Big), c\hat{A}_t\Big)\Big], \tag{1}$$

where $c > 1$ indicates the lower bound, and $\epsilon$ is the original clip in PPO.

**Value updates**. To decrease the variance of value estimation, similar to [33], we use full information about the game state, including observations hidden from the policy, as input to the value function. Note that this is performed only during training, as we only use the policy network during evaluation. In order to estimate the value of the ever-changing game state more accurately, we introduce multi-head value (MHV) into MOBA by decomposing the reward, which is inspired by the hybrid reward architecture (HRA) used on the Atari game Ms. Pac-Man [32]. Specifically, we design five reward categories as the five value heads, as shown in Fig. 1, based on game expert's knowledge and the accumulative value loss in each head. These value heads and the reward items contained in each head are: 1) Farming related: gold, experience, mana, attack monster, no-op (not acting); 2) KDA related: kill, death, assist, tyrant buff, overlord buff, expose invisible enemy, last hit; 3) Damage related: health point, hurt to hero; 4) Pushing related: attack turrets, attack enemy home base; 5) Win/lose related: destroy enemy home base.

$$\mathcal{L}^{value}(\theta) = \hat{\mathbb{E}}_t\Big[\sum_{head_k}(R_t^k - \hat{V}_t^k)^2\Big], \quad \hat{V}_t = \sum_{head_k} w_k \hat{V}_t^k, \tag{2}$$

where $R_t^k$ and $\hat{V}_t^k$ are the discounted reward sum and value estimation of the $k^{th}$ head, respectively. Then, the total value estimation is the weighted sum of the head value estimates.

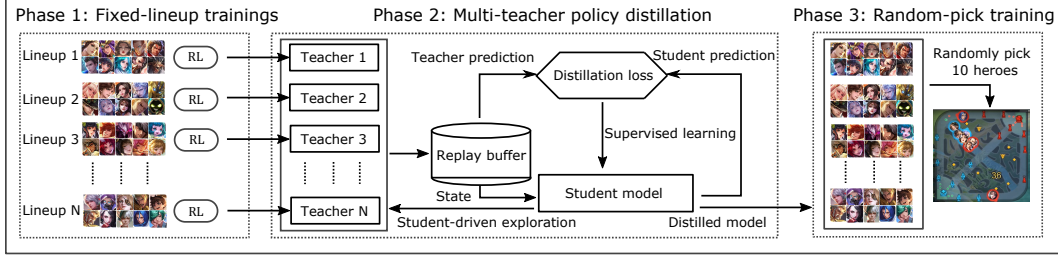

Figure 2: The flow of curriculum self-play learning: **1)** Small task with small model. We divide heroes into groups, and start with training fixed lineups, i.e., 5 fixed heroes VS another 5 fixed heroes, via self-play RL. **2)** Distillation. We adopt multi-teacher policy distillation. **3)** Continued learning.

## 3.3 Multi-agent Training

As discussed, large hero pool leads to a huge number of lineups. When using self-play reinforcement learning, the 10 agents playing one MOBA game is faced with a non-stationary moving-target problem [4, 13]. Furthermore, the lineup varies from one self-play to another, making policy learning even more difficult. Presenting disordered agent combinations for training leads to degraded performance [24]. This calls for a paradigm to guide agents learning in MOBA.

Inspired by the idea of curriculum learning [1], i.e., machine learning models can perform better when the training instances are not randomly presented but organized in a meaningful order which illustrates gradually more concepts, we propose curriculum self-play learning (CSPL) to guide MOBA AI learning. CSPL includes three phases, shown in Fig. 2, described as follows. The rule of advancing to the next phase in CSPL is based on the convergence of Elo scores.

In Phase 1, we start with easy tasks by training fixed lineups. To be specific, in the 40-hero case, we divide the heroes to obtain four 10-hero groups. The self-play is performed separately for each group. The 10-hero grouping is based on the balance of two 5-hero teams, with a win-rate close to 50% to each other. The win-rate of lineups can be obtained from the vast amount of human player data. We select balanced teams because it is practically effective to policy improvement in self-play [33, 15]. To train teachers, we use a smaller model with almost half parameters of the final model in Phase 3, which will be detailed in Section 4.1.

In Phase 2, we focus on how to inherit the knowledge mastered by the fixed-lineup self-plays. Specifically, we apply multi-teacher policy distillation [26], using models from Phase 1 as teacher models ($\pi$), which are merged into a single student model ($\pi_\theta$). The distillation is a supervised process, based on the loss function in Eq. 3, where $H^\times(p(s)||q(s))$ denotes Shannon's cross entropy between two distributions over actions $-E_{a\sim p(s)}[\log q(a|s)]$, $q_\theta$ is the sampling policy, $\hat{V}^{(k)}(s)$ is the value function, and $head_k$ denotes the $k$-th value head mentioned in the previous section.

$$\mathcal{L}^{distil}(\theta) = \sum_{teacher_i} \hat{\mathbb{E}}_{\pi_\theta}[\sum_t H^\times(\pi_i(s_t)||\pi_\theta(s_t)) + \sum_{head_k} (\hat{V}_i^k(s_t) - \hat{V}_\theta^k(s_t))^2]. \tag{3}$$

With the loss of cross entropy and mean square error of value predictions, we sum up these losses from all the teachers. As a result, the student model distills both of policy and value knowledge from the fixed-lineup teachers. During distillation, the student model is used for exploration in the fixed-lineup environments where teachers are trained, known as student-driven policy distillation [9]. The exploration outputs actions, states and the teacher's predictions (used as guidance signal for supervised learning) into the replay buffer.

In Phase 3, we perform continued training by randomly picking lineups in the hero pool, using the distilled model from Phase 2 for model initialization.

## 3.4 Learning to draft

An emerging problem brought by expanding the hero pool is drafting, a.k.a. hero pick or hero selection. Before a MOBA match starts, two teams go through the process of picking heroes, which

directly affects future strategies and match result. Given a large hero pool, e.g., 40 heroes (more than $10^{11}$ combinations), a complete tree search method like the Minimax algorithm used in *OpenAI Five* [2] will be computationally intractable [5].

To manage this, we develop a drafting agent leveraging Monte-Carlo tree search (MCTS) [7] and neural networks. MCTS estimates the long-term value of each pick, and the hero with the maximum value will be picked. The particular MCTS version we use is Upper Confidence bounds applied to Trees (UCT) [19]. When drafting, a search tree is built iteratively, with each node representing the state (which heroes have been picked by both teams) and each edge representing the action (pick a hero that has not been picked) resulting to a next state.

The search tree is updated based on the four steps of MCTS for each iteration, i.e., selection, expansion, simulation, and backpropagation, during which the simulation step is the most time-consuming. To speed up simulation, different from [5], we build a value network to predict the value of the current state directly instead of the inefficient random roll-out to get the reward for backpropagation, which is similar to AlphaGo Zero [31]. The training data of the value network is collected via a simulated drafting process played by two drafting strategies based on MCTS. When training the value network, Monte-Carlo roll-out is still performed until reaching the terminal state, i.e., the end of the simulated drafting process. Note that, for board games like Chess and Go, the terminal state determines the winner of the match. However, the end of the drafting process is not the end of a MOBA match, so we cannot get match results directly. To deal with this, we first build a match dataset via self-play using the RL model trained in Section 3.3, and then we train a neural predictor for predicting the win-rate of a particular lineup. The predicted win-rate of the terminal state is used as the supervision signal for training the value network. The architectures of the value network and the win-rate predictor are two separate 3-layer MLPs. For the win-rate predictor, the input feature is the one-hot representation of the 10 heroes in a lineup, and the output is the win-rate ranged from 0 to 1. For the value network, the input representation is the game state of the current lineup, containing one-hot indexes of picked heroes in the two teams, default indexes of unpicked heroes, and the index of the team which is currently picking, while the output is the value of the state. On the other hand, the selection, expansion and backpropagation steps in our implementation are the same as the normal MCTS [19, 5].

### 3.5 Infrastructure

To manage the variance of stochastic gradients introduced by MOBA agents, we develop a scalable and loosely-coupled infrastructure to construct the utility of data parallelism. Specifically, our infrastructure follows the classic Actor-Learner design pattern [12]. Our policy is trained on the Learner using GPUs while the self-play happens on the Actor using CPUs. The experiences, containing sequences of observations, actions, rewards, etc., are passed asynchronously from the Actor to a local replay buffer on the Learner. Significant efforts are dedicated to improving the system throughput, e.g., the design of transmission mediators between CPUs and GPUs, the IO cost reduction on GPUs, which are similar to [38]. Different from [38], we further develop a centralized inference module on the GPU side to optimize resource utilization, similar to the Learner design in a recent infrastructure called Seed RL [11].

## 4 Evaluation

### 4.1 Experimental Setup

We test on *Honor of Kings*, which is the most popular MOBA game worldwide and has been actively used as the testbed for recent AI advances [10, 35, 16, 36, 38, 37].

Our RL infrastructure runs over a physical computing cluster. To train our AI model, we use 320 GPUs and 35,000 CPUs, referred to as `one resource unit`. For each of the experiments conducted below, including ablation study, time and performance comparisons, we use the same quantity of resources to train, i.e., one resource unit, unless otherwise stated. Our cluster can support 6 to 7 such experiments in parallel. The mini-batch size per GPU card is 8192. We develop 9,227 scalar features containing observable unit attributions and in-game stats, and 6 channels of spatial features read from the game engine with resolution 6*17*17. Each of the teacher models has 9 million parameters, while the final model has 17 million parameters. LSTM unit sizes for teacher and final models are 512 and 1024, respectively. LSTM time step is 16 for all models. For teacher models, we train using

half resource unit, since they are relatively small-sized. To optimize, we use Adam [17] with initial learning rate 0.0001. For Dual-clip PPO, the two clipping hyperparameters $\epsilon$ and $c$ are set as 0.2 and 3, respectively. The discount factor is set as 0.998. We use generalized advantage estimation (GAE) [27] for reward calculation, with $\lambda = 0.95$ to reduce the variance caused by delayed effects.

For drafting, the win-rate predictor is trained with a match dataset of 30 million samples. These samples are generated via self-play using our converged RL model trained from CSPL. And the value network is trained using 100 million samples (containing 10 million lineups; each lineup has 10 samples because we pick 10 heroes for a completed lineup) generated from MCTS-based drafting strategies. The labels for the 10 samples in each lineup are the same, which is calculated using the win-rate predictor.

To evaluate the trained AI's performance, we deploy the AI model into *Honor of Kings* to play against top human players. For online use, the response time of AI is 193 ms, including the observation delay (133 ms) and reaction delay (about 60 ms), which is made up of processing time of feature, model, result, and the network delay. We also measure the APM (action per minute) of AI and top human players. The averaged APMs of our AI and top players are comparable (80.5 and 80.3, respectively). The proportions of high APMs (APM $\geq$ 300 for *Honor of Kings*) during games are 4% for top players and 5% for our AI, respectively. We use the Elo rating [8] for comparing different versions of AI, similar to other Game AI programs [30, 33].

## 4.2 Experimental Results

### 4.2.1 AI Performance

We train an AI for playing a pool of 40 heroes [4] in *Honor of Kings*, covering all hero roles (tank, marksman, mage, support, assassin, warrior). The scale of hero pool is 2.4x larger than previous MOBA AI work [2], leading to $2.1 \times 10^{11}$ more agent combinations. During the drafting phase, human players can pick heroes from the 40-hero pool. When the match starts, there are no restrictions to game rules, e.g., players are free to build any item or use any summoner ability they prefer.

We invite professional esports players of *Honor of Kings* to play against our AI. From Feb. 13th, 2020 to Apr. 30th, 2020, we conduct weekly matches between AI and current professional esports teams. The professionals were encouraged to use their skilled heroes and to try different team strategies. In a span of 10-week's time, a total number of 42 matches were played. Our AI won 40 matches of them (win rate 95.2%, with confidence interval (CI) [22] [0.838, 0.994]). By comparison, professional tests conducted by other related Game AI systems are: 11 matches for *AlphaStar Final* (10 win 1 lose, CI [0.587, 0.997]), and 8 matches for *OpenAI Five* (8 win 0 lose, CI [0.631, 1]). A number of episodes and complete games played between AI and professionals are publicly available at: `https://sourl.cn/NVwV6L`, in which various aspects of the AI are shown, including long-term planning, macro-strategy, team cooperation, high-level turret pushing without minions, solo competition, counter strategy to enemy's gank, etc. Through these game videos, one can clearly see the strategies and micro controls mastered by our AI.

From May 1st, 2020 to May 5th, 2020, we deployed our AI into the official release of *Honor of Kings* (Version 1.53.1.22, released on Apr. 29th, 2020; AI-game entry switch-on at 00:00:00, May 1st), to play against the public. To rigorously evaluate whether our AI can counter diverse high-level strategies, only top ranked human players (at about the level of High King in *Honor of Kings*) are allowed to participate, and participants can play repeatedly. To encourage engagement, players who can defeat the AI will be given a honorary title and game rewards in *Honor of Kings*. As a result, our AI was tested against top human players for 642,047 matches. AI won 627,280 of these matches, leading to a win-rate of 97.7% with confidence interval [0.9766, 0.9774]. By comparison, public tests from the final version of *AlphaStar* and *OpenAI Five* are 90 matches and 7,257 matches, respectively, with no requirements of game level to the participated human players.

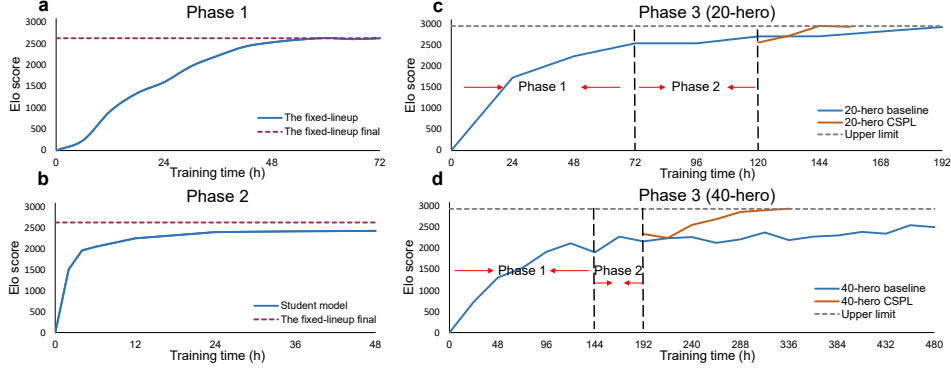

Figure 3: The training process: **a)** The training of a teacher model, i.e., Phase 1 of CSPL. **b)** The Elo change during distillation, i.e., Phase 2 of CSPL. The student's converged Elo is marginally below the teacher. **c)** and **d)** Compare the Elo change of CSPL with the baseline method, for the 20-hero and 40-hero cases, respectively. Note that the baseline has no Phase 1 and Phase 2. CSPL has better scalability than the baseline when expanding the hero pool.

Table 1: Comparing training time of CSPL and the baseline.

| #Heroes | Training method | Phase1 | Phase2 | Phase3 | Total time | Note |
|---------|-----------------|--------|--------|--------|------------|------|
| 20 | Baseline | 0 | 0 | 192 h | 192 h | slow convergence |
| 20 | CSPL | 72 h | 48 h | 24 h | 144 h | fast convergence |
| 40 | Baseline | 0 | 0 | NA (> 480 h) | > 480 h | very slow or non convergence |
| 40 | CSPL | 144 h | 48 h | 144 h | 336 h | fast convergence |

Prior to supporting 40 heroes, we trained a 20-hero [5] AI version, using the same training paradigm. The AI was tested against professional teams for 30 matches, and AI dominated the matches with 100% win rate (30:0). We are in the process of training a complete pool of heroes (there are 101 heroes in total in *Honor of Kings* as per Oct. 2020) using our paradigm.

### 4.2.2 Training Process

In this section, we compare the training process between our paradigm and OpenAI's method [2], i.e., randomly picking heroes, when trained with different hero pool sizes, including 20-hero and 40-hero.

Both *OpenAI Five* [2] and our development experience confirm that training with the minimal hero pool (10-hero) is able to reach professional esports player's level rapidly, whose Elo score is thereby used as the upper limit criterion in our experiments. With this criterion, we can evaluate the detailed learning process of different sizes of hero pools using the two methods. The 10 heroes chosen are also included in the 20- and 40-hero pools. Note that the 10-hero lineup used for computing upper limit Elo score does not have an overlap with the lineups for preparing teacher models during the Phase 1 of CSPL, for fairness of evaluation.

In Fig. 3, we illustrate the whole training process of CSPL and the baseline. And in Table 1, we compare the concrete training time between the two methods. Specifically, the Elo growing trend of a teacher model is shown in Fig. 3-a., from which we observe that the Elo score grows with the training length and becomes relatively steady after about 72 hours. As mentioned, each teacher is trained with half resource unit. Therefore, one resource unit is used to train two teacher models in parallel in Phase 1 in practice. Fig. 3-b shows the policy distillation process, from which we can see that the AI ability rapidly increases in the beginning, and soon converges to an Elo score that is marginally below the teacher's score. We empirically observe that the convergence in student-driven policy distillation is not sensitive to the hero pool size. Both 20-hero and 40-hero cases can converge rapidly within two days. Fig. 3-c and Fig. 3-d provide a direct comparison between the two training processes. For 20-hero, we see that both CSPL and the baseline converge within a reasonable amount of time, while

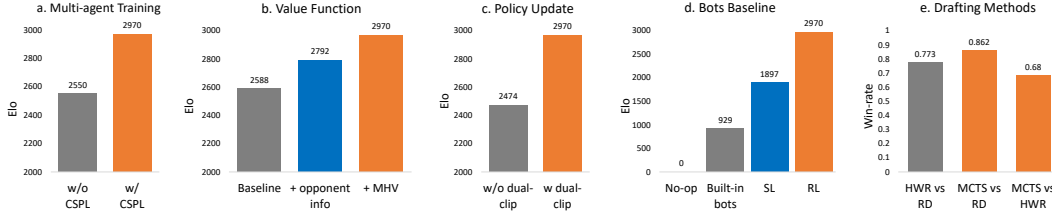

Figure 4: Ablations for key components: **a**) comparing training methods using Elo (with CSPL and without CSPL); **b**) comparing different compositions of value functions using Elo; **c**) comparing ways of policy updates using Elo; **d**) Elo scores of built-in bots and supervised learning agents using human game data and the final reinforcement learning agents; **e**) comparing averaged win-rates when using different drafting methods (RD: picking heroes randomly, HWR: picking heroes with highest win-rates, MCTS: picking heroes with our drafting method).

CSPL can converge faster than the baseline. However, for 40-hero, the baseline method results in unacceptably slow training, which fails to converge, with an Elo score 400 lower than the upper limit even being trained for 480 hours. By comparison, CSPL converges much faster using a total amount of training time of about 336 hours. Overall, training MOBA AI playing a large hero pool benefits significantly from our proposed method.

### 4.2.3 Ablations

To further analyze the components in our method, we have run several internal ablations, as shown in Fig. 4. All these comparative experiments are carried out based on the final 40-hero AI version with equal training time and resource. We observe that, for multi-agent training, training with CSPL is helpful to the final AI performance (Fig. 4-a). For value estimation, using invisible opponent information and multi-head value can both increase the Elo score effectively (Fig. 4-b), when compared with the baseline value estimation method in *OpenAI Five* [2]. Fig. 4-c indicates that applying dual-clip PPO is beneficial to the ability of our agents, due to its training stability, as analyzed in Section 3.2. In Fig. 4-d, the Elo scores of built-in bots, supervised learning agents and the final reinforcement learning agents are provided. Ratings are anchored by a bot that never acts. Finally, drafting methods are compared in Fig. 4-e. The number of matches played for each pair of drafting strategies is 1000. We can see that our proposed drafting method (MCTS-based) outperforms RD (each time, sample an unpicked hero **r**and**o**mly) and HWR (pick the hero with **h**ighest **w**in-**r**ates based on frequency counts), with 0.86 and 0.68 mean win-rates, respectively.

## 5  Conclusion and Future Work

In this paper, we proposed a MOBA AI learning paradigm towards playing full MOBA games with deep reinforcement learning. We developed a combination of novel and existing learning techniques, including off-policy adaption, multi-head value estimation, curriculum self-play learning, multi-teacher policy distillation, and Monte-Carlo tree search, to deal with the emerging problems caused by training and playing a large pool of heroes. Tested on *Honor of Kings*, the most popular MOBA game at present, our AI can defeat top human players with statistical significance. To the best of our knowledge, this is the first reinforcement learning based MOBA AI program that can play a pool of 40 heroes and more. Furthermore, there has been no AI program for sophisticated strategy video games working with our large-scale, rigorous, and repeated performance testing.

In the future, we will continue to work on complete hero pool support, and to investigate more efficient training methods to further shorten the MOBA AI learning process. To facilitate research on game intelligence, we will also develop subtasks of MOBA-game-playing for the AI community.

# 6 Broader Impact

**To the research community**. MOBA (Multiplayer Online Battle Arena) poses a grand challenge to the AI community. We would believe that mastering a typical MOBA game without restrictions will become the next AI milestone like *AlphaGo* or *AlphaStar*. To this end, this paper is introducing a MOBA AI learning paradigm towards this goal. Moreover, the proposed methodology is based on general-purpose machine learning components that are applicable to other similar multiplayer domains. Our results suggest that curriculum-guided reinforcement learning can help handle very complex tasks involving multi-agent competition and cooperation, real-time decision-making, imperfect observation, complex strategy space, and combinatorial action space. We herewith expect this work to provide inspirations to other complex real-world problems, e.g., real-time decisions of robotics.

**To the game industry**. *Honor of Kings*, published by Tencent, has a tremendously large user group. It was reported to be the world's most popular and highest-grossing game of all time, as well as the most downloaded App worldwide [6]. Our AI has found several real-world applications in the game, and is changing the way that MOBA game designers work, particularly game balance designers, elaborated as follows: 1) Game balance testing. In MOBA and many other game types, balancing the ability of each character is essential. The numerical values and skill-sets design of a MOBA hero, e.g., the physical/magical attack value, blood, and skill types, are traditionally set based on the experience of game balance designers. Value adjustments to a hero must be tested in the Beta game servers for months, to see its win-rate in the hero pool through a significantly large number of human matches. In self-play reinforcement learning, the agents are very sensitive to feature changes and hero adjustments affect the win-rate of the team. Using similar techniques presented in this paper, we have constructed a balance testing tool for *Honor of Kings*. 2) PVE (player vs environment) game mode. We had deployed earlier checkpoints trained by our method (with a much weaker ability than the AI in this paper, mainly for entertainment) into *Honor of Kings*. These checkpoints are for players at all levels. 3) AFK hosting. It happens that players in casual matches drop offline or AFK (away from the keyboard) during a game due to an unstable network, temporary emergencies, etc. We have developed a preliminary learning-based AI to host the dropped players.

**To the esports community**. The playing style of our AI is different from normal playing of human esports players [7]. For example, in MOBA, a commonly seen opening-strategy for human teams is the three-lane-strategy, i.e., marksman and warrior heroes go to bottom and top lanes, respectively, while the mage hero plays the middle. However, such a strategy has seldom been adopted by the AI team. Our AI has its way of fast upgrading and team cooperation, which has inspired new strategies to professional players. As commented by a professional coach, "AI's resource allocation is a bit weird but effective. After trying out some of AI's playing style, we observe a slight increase of gold and experience gained during certain game phases. Another finding given by AI is that some marksman heroes are suitable to play middle, apart from playing bottom. These are interesting and helpful to us." In the future, we would believe that with larger hero pool support, the strategies explored by self-play reinforcement learning will have an even broader impact on how esports players play the game.

# 7 Funding Disclosure

This work is supported by the Tencent AI Department and the Tencent TiMi Studio.

## Footnotes

[1]In this paper, MOBA refers to the standard MOBA 5v5 game, unless otherwise stated.

[2]*OpenAI Five* has two limitations from the regular game: 1) the major one is the limit of hero pool, i.e., only a subset of 17 heroes supported; 2) some game rules were simplified, e.g., certain items were not allowed to buy.

[3]In *Honor of Kings*, a player's game level can be: No-rank, Bronze, Silver, Gold, Platinum, Diamond, Heavenly, King (a.k.a, Champion) and High King (the highest), in ascending order.

[4]40-hero pool: *Di Renjie, Consort Yu, Marco Polo, Lady Sun, Gongsun Li, Li Yuanfang, Musashi Miyamoto, Athena, Luna, Nakoruru, Li Bai, Zhao Yun, Wukong, Zhu Bajie, Wang Zhaojun, Wu Zetian, Mai Shiranui, Diaochan, Gan&Mo, Shangguan Wan'er, Zhang Liang, Cao Cao, Xiahou Dun, Kai, Dharma, Yao, Ma Chao, Ukyo Tachibana, Magnus, Hua Mulan, Guan Yu, Zhang Fei, Toro, Dong Huang Taiyi, Zhong Kui, Su Lie, Taiyi Zhenren, Liu Shan, Sun Bin, Guiguzi.*

[5]20-hero pool: *Athena, Consort Yu, Wu Zetian, Zhang Fei, Cao Cao, Nakoruru, Di Renjie, Wang Zhaojun, Dharma, Toro, Luna, Marco Polo, Diaochan, Sun Bin, Kai, Li Bai, Gongsun Li, Mai Shiranui, Magnus, Guiguzi.*

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
