[Supplementary Material]

# 1 Supplementary Materials

## 1.1 Infrastructure Design

In Fig. 1, we show our infrastructure, called *KaiWu*. It consists of four major components: AI Server, Inference Server, RL Learner and Memory Pool. The AI Server (the Actor) covers the interaction logic between the agents and environment. The Inference Server is for centralized batch inference on the GPU side. The RL Learner (the Learner) is a distributed training environment for RL model training. And the Memory Pool is for storing experience replay, implemented as a memory-efficient circular queue. The website of our infrastructure is: `aiarena.tencent.com`.

Figure 1: Our infrastructure design.

We used a large amount of computing resources for building our AI, due to the complex nature of the problem we study. In fact, the computing resources required for complex game-playing AI programs are non-trivial, e.g., AlphaGo Lee Sedol version (280 GPUs), OpenAI Five Final (1920 GPUs), and the final version of AlphaStar (3072 TPUv3 cores). We will continue to work on the infrastructure efficiency to further reduce the computational cost.

## 1.2 Game Environment

In Fig. 2, we show a game UI of *Honor of Kings*. All the experiments in the paper were carried out using a fixed big version (Version 1.53 series) of game core of *Honor of Kings* for fair comparison.

Figure 2: Game UI of *Honor of Kings*. The hero controlled by the player is called "main hero". Bottom-left is the movement controller (C.1), while the right-bottom set of buttons are ability controllers (C.2, C.3). Players can observe game situations via the screen (local view), obtain game states with dashboard (B), and obtain a global view with the top-left mini map (A).

## 1.3 Hero Pool

The hero names of the 40-hero pool are as follows:

*Di Renjie, Consort Yu, Marco Polo, Lady Sun, Gongsun Li, Li Yuanfang, Musashi Miyamoto, Athena, Luna, Nakoruru, Li Bai, Zhao Yun, Wukong, Zhu Bajie, Wang Zhaojun, Wu Zetian, Mai Shiranui, Diaochan, Gan&Mo, Shangguan Wan'er, Zhang Liang, Cao Cao, Xiahou Dun, Kai, Dharma, Yao,*

*Ma Chao, Ukyo Tachibana, Magnus, Hua Mulan, Guan Yu, Zhang Fei, Toro, Dong Huang Taiyi, Zhong Kui, Su Lie, Taiyi Zhenren, Liu Shan, Sun Bin, Guiguzi.*

The 10-hero pool used for constructing the evaluation criteria includes the following heroes:

*Athena, Luna, Marco Polo, Di Renjie, Zhang Fei, Sun Bin, Wang Zhaojun, Diaochan, Kai, Cao Cao.*

As mentioned, in Phase 1 of our CSPL, we divide the heroes into several subgroups. The fixed-lineups used for Phase 1 in CSPL are summarized in Table 1.

Table 1: Fixed lineups for 20 heroes and 40 heroes.

| Pool | Fixed lineups for teacher models (10 heroes each line, the left 5 vs the right 5 heroes). |
|------|---------------------------------------------------------------------------------------------|
| 20 | Athena, Consort Yu, Wu Zetian, Zhang Fei, Cao Cao, Nakoruru, Di Renjie, Wang Zhaojun, Dharma, Toro <br> Luna, Marco Polo, Diaochan, Sun Bin, Kai ,Li Bai, Gongsun Li, Mai Shiranui, Magnus, Guiguzi |
| 40 | Athena, Consort Yu, Wu Zetian, Zhang Fei, Cao Cao, Nakoruru, Di Renjie, Wang Zhaojun, Dharma, Toro <br> Luna, Marco Polo, Diaochan, Sun Bin, Kai ,Li Bai, Gongsun Li, Mai Shiranui, Magnus, Guiguzi <br> Musashi Miyamoto, Lady Sun, Gan&Mo, Dong Huang Taiyi, Xiahou Dun, Zhao Yun, Li Yuanfang, Shangguan Wan'er, Ma Chao ,Su Lie <br> Zhu Bajie ,Zhang Liang, Ukyo Tachibana, Hua Mulan, Taiyi Zhenren, Guan Yu, Zhong Kui, Wukong, Liu Shan, Yao |

## 1.4 Agent Action

Table 2 provides the details of our action space design.

Table 2: Agent action space.

| Action | Detail | Description |
|--------|--------|-------------|
| What | Illegal action | Placeholder. |
| | None action | Executing nothing or stopping continuous action. |
| | Move | Moving to a certain direction determined by move x and move y. |
| | Normal Attack | Executing normal attack to an enemy unit. |
| | Skill1 | Executing the first skill. |
| | Skill2 | Executing the second skill. |
| | Skill3 | Executing the third skill. |
| | Skill4 | Executing the fourth skill (only a few heroes have Skill4). |
| | Summoner ability | An additional skill choosing before the game begins (10 to choose). |
| | Return home(Recall) | Returning to spring, should be continuously executed. |
| | Item skill | Some items can enable an additional skill to player's hero. |
| | Restore | Blood recovering continuously in 10s, can be disturbed. |
| | Collaborative skill | Skill given by special ally heroes. |
| How | Move X | The x-axis offset of moving direction. |
| | Move Y | The y-axis offset of moving direction. |
| | Skill X | The x-axis offset of a skill. |
| | Skill Y | The y-axis offset of a skill. |
| Who | Target unit | The game unit(s) chosen to attack. |

## 1.5 Reward Design

Table 3 shows the details of our reward.

## 1.6 Feature Design

The detailed features extracted by our model are listed in Table 4. The feature consists of two main types: scalar features and spatial features. Scalar features include unit attributes, in-game statistics and invisible opponent information. Note that the invisible opponent information is only applied to the value network during training.

The way of feature normalization is as follows. For continuous features, we use their maximum and minimum values to normalize them into the interval of [0,1], such as health point (HP), mana, speed, etc. For example, the HP of a hero is normalized to a value between 0 (death) and 1 (full health). And for discrete features, we use one-hot representation by enumerating all possible values, such as skill level, kill-death-assist stats, etc. For example, the in-game skill level of a hero can be level 1 to level 15, so we use a one-hot vector of dimension 15 for representation.

Table 3: Reward design details.

| Head | Reward Item | Weight | Type | Description |
|---|---|---|---|---|
| Farming Related | Gold | 0.005 | Dense | The gold gained. |
| | Experience | 0.001 | Dense | The experience gained. |
| | Mana | 0.05 | Dense | The rate of mana (to the fourth power). |
| | No-op | -0.00001 | Dense | Stop and do nothing. |
| | Attack monster | 0.1 | Sparse | Attack monster. |
| KDA Related | Kill | 1 | Sparse | Kill a enemy hero. |
| | Death | -1 | Sparse | Being killed. |
| | Assist | 1 | Sparse | Assists. |
| | Tyrant buff | 1 | Sparse | Get buff of killing tyrant, dark tyrant, storm tyrant. |
| | Overlord buff | 1.5 | Sparse | Get buff of killing the overlord. |
| | Expose invisible enemy | 0.3 | Sparse | Get visions of enemy heroes. |
| | Last hit | 0.2 | Sparse | Last hitting an enemy minion. |
| Damage Related | Health point | 3 | Dense | The health point of the hero (to the fourth power). |
| | Hurt to hero | 0.3 | Sparse | Attack enemy heroes. |
| Pushing Related | Attack turrets | 1 | Sparse | Attack turrets. |
| | Attack crystal | 1 | Sparse | Attack enemy home base. |
| Win/Lose Related | Destroy home base | 2.5 | Sparse | Destroy enemy home base. |

Table 4: Feature details.

| Feature Class | Field | Description | Dimension |
|---|---|---|---|
| **1. Unit feature** | Scalar | Includes heroes, minions, monsters, and turrets | 8599 |
| Heroes | Status | Current HP, mana, speed, level, gold, KDA, buff, bad states, orientation, visibility, etc. | 1842 |
| | Position | Current 2D coordinates | 20 |
| | Attribute | Is main hero or not, hero ID, camp (team), job, physical attack and defense, magical attack and defense, etc. | 1330 |
| | Skills | Skill 1 to Skill N's cool down time, usability, level, range, buff effects, bad effects, etc. | 2095 |
| | Item | Current item lists | 60 |
| Minions | Status | Current HP, speed, visibility, killing income, etc. | 1160 |
| | Position | Current 2D coordinates | 80 |
| | Attribute | Camp (team) | 80 |
| | Type | Type of minions (melee creep, ranged creep, siege creep, super creep, etc.) | 200 |
| Monsters | Status | Current HP, speed, visibility, killing income, etc. | 868 |
| | Position | Current 2D coordinates | 56 |
| | Type | Type of monsters (normal, blue, red, tyrant, overlord, etc.) | 168 |
| Turrets | Status | Current HP, locked targets, attack speed, etc. | 520 |
| | Position | Current 2D coordinates | 40 |
| | Type | Type of turrets (tower, high tower, crystal, etc.) | 80 |
| **2. In-game stats feature** | Scalar | Real-time statistics of the game | 68 |
| Static statistics | Time | Current game time | 5 |
| | Gold | Golds of two camps | 12 |
| | Alive heroes | Number of alive heroes of two camps | 10 |
| | Kill | Kill number of each camp | 6 |
| | Alive turrets | Number of alive turrets of two camps | 8 |
| Comparative statistics | Gold diff | Gold difference between two camps | 5 |
| | Alive heroes diff | Alive heroes difference between two camps | 11 |
| | Kill diff | Kill difference between two camps | 5 |
| | Alive turrets diff | Alive turrets difference between two camps | 6 |
| **3. Invisible opponent information** | Scalar | Invisible information used for the value net | 560 |
| Opponent heroes | Position | Current 2D coordinates, distances, etc. | 120 |
| NPC | Position | Current 2D coordinates of all non-player characters, including minions, monsters, and turrets | 440 |
| **4. Spatial feature** | Spatial | 2D image-like, extracted in channels for convolution | 6x17x17 |
| Skills | Region | Potential damage regions of ally and enemy skills | 2x17x17 |
| | Bullet | Bullets of ally and enemy skills | 2x17x17 |
| Obstacles | Region | Forbidden region for heroes to move | 1x17x17 |
| Bushes | Region | Bush region for heroes to hide | 1x17x17 |