[Reviews · NeurIPS 2020]

Review 1

Summary and Contributions: This paper proposes a MOBA AI learning paradigm that is able to train agents playing full MOBA games with deep reinforcement learning. The method applying a variety of learning techniques, including multi-head value estimation, curriculum self-play learning, policy distillation, and Monte-Carlo tree-search, in training and playing a large pool of heroes, meanwhile addressing the scalability issue. The agents were tested on Honor of Kings, a popular MOBA game. The win rates against professional players and top-amateur players were 95.2% (40 out of 42 games) and 97.7% (642,047 out of 627,280). The paper provides an AI learning method to develop AI agents conducting game balance testing and strategy exploration, which are helpful to improve player experience and improving players’ strategies, on MOBA games.

Strengths: 1. Value Update: In section 3.2, although the multi-head value concept has existed for years, the reward items (e.g., farming related, KDA related, Damage related and Pushing related) used in this work still provide insight into reward design for other MOBA games or video games. 2. Curriculum Self-Play Learning (CSPL): In section 3.3, the method of combining curriculum learning and student-driven policy distillation that stabilizes the training and overcomes the problem of a huge number of lineups is novel and effective according to the experimental results. 3. Learning to Draft: The paper points out the drawbacks of using minimax algorithm and Monte-Carlo Tree Search (MCTS) without value network. In order to improve the efficiency, this work develops a method leveraging MCTS and neural network based on AlphaZero. 4. In general, the MOBA AI learning paradigm proposed in this work provides a way to train AI agents up to top-professional player strength, namely the win rates against professional players and top-amateur players were 95.2% (40 out of 42 games) and 97.7% (642,047 out of 627,280). Also, it has the potential to become a game balance testing tool to improve the game quality and player experience and to explore different playing styles and strategies to help human players improve their skills.

Weaknesses: Some presentation described in “clarity”. BTW, the training method proposed in this work needs a massive amount of computing resources to train AI agents, for example, this paper refers to 320 GPUs and 35,000 CPUs as one resource unit, which is infeasible to most of the researchers and research units. Any more discussion about this issue would be better.

Correctness: I don’t find significant errors in this paper.

Clarity: Pros: 1. The description of the method is pretty clear and easy to understand. This paper points out the methods it applies and the improvements it develops to solve the problems. 2. The specification of the computing cluster and the setting for each experiment are clear and detailed. Cons: 1. In section 3.3, The rule of advancing to the next curriculum phase is unclear. In the experiment in Table 1, the training time for each phase is not proportional to the pool size (both trainings in 20-heroes and 40-heroes take 48 hours for phase 2). 2. In section 3.4 Learning to draft, the paper only provides the training process and network structure roughly (3-layer MLP), and the details of the networks and training hyperparameters of value network and win-rate predictor were not provided. 3. In section 3.2 of supplementary material, the paper provides the features for training, but the range of each feature was unclear. It would cause difficulties when readers are trying to reproduce this work.

Relation to Prior Work: In section 3.4, the paper describes how to improve the methods, such as minimax algorithm and MCTS without value network, to speed up the process of searching for the best heroes to pick.

Reproducibility: No

Additional Feedback: Reproducibility problem: 1. In section 3.4, the paper only provides the training process and network structure roughly (3-layer MLP) and the details of the networks and training hyperparameters of value network and win-rate predictor were not provided. 2. In section 3.2 of supplementary material, the paper provides the features for training, but the range of each feature was unclear. Since the training in reinforcement learning is sensitive to the range of input, the range of features should be provided to make the work reproducible. Typos: 1. Line 224: Monte-Carole -> Monte-Carlo 2. Line 380: wired -> weird After reading the rebuttal, I think some description is still insufficient, e.g., how MCTS works is still unclear, how the MCTS drafting match dataset was collected, how they decide the number 1600 for iteration count. Since this technique is one of the major contributions in this paper, the authors should provide more details of implementing and training for reproducing this work. So, I would like to adjust the overall score for this submission to 6.


Review 2

Summary and Contributions: The paper describes an ML framework to achieve expert playing strength in a full MOBA game by combining and refining existing methods for off-policy adaption, multi-head value estimation, curriculum self-play learning, multi-teacher policy distillation, and applying Monte-Carlo tree search for the combinatorial hero selection phase. Putting everything together and training the system on a massive computing platform resulted in an expert MOBA player.

Strengths: The paper enhances and integrates a lot of building blocks to create a high-performance system that plays a popular video game well, similar to AlphaGo, OpenAI-Five, and AlphaStar. The employed method improvements are relevant to the NeurIPS community. Tthe final large scale experimental evaluation is nice.

Weaknesses: - the individual contributions seem to be incremental in nature, with the majority of building blocks having been reported in the literature already. Examples include curriculum learning, MCTS vs. AlphaBeta, and off-policy adaption - it hasn't been made clear in what way the developed methods are general enough to tackle other problems. A second application domain would help. - the MCTS method for drafting lacks details. As described, it can't be reproduced - using huge amounts of computation for training also makes it hard to reproduce this work. Ideally, the authors would publish their AI system for the AI community to analyse - the non-CSPL experiments haven't been run long enough to make a clear call that they won't achieve similar ELO values - what do the learning curves actually depict? one-shot? average? cherry-picked? If average - what are the confidence bounds? - computation time should be reported as petaflop/s hours and iteration numbers to make them comparable with other related work - listing xyz hours isn't helpful

Correctness: The paper results are heuristic in nature. There are no formal theorems or proofs. The experimental methodology is valid, although I would have loved to see more games being played against professional teams and also giving them the opportunity to learn to model the AI system to exploit weaknesses.

Clarity: The paper suffers from numerous grammar and phrasing problems. It needs to be proofread thoroughly. Some details are listed below.

Relation to Prior Work: The paper describes related work well and makes clear what the prior contributions are that their work is based on.

Reproducibility: No

Additional Feedback: some details: p1 - define MOBA in the abstract - such as multi-agent, grammar - attention already reference(s)? - raw game complexity what is this? - Dota => DOTA - 17 heroes 18 I believe. - issue skillfully self-praise - top[-] professional and top[-] amateur the latter seems to be implied by the former - esports => E-sports - l30: state space or action space sizes does not imply decision complexity p2 - disordered hero combination not sure what this means - following [that of] [39] - for neural network[s] - l64: minimax algorithm I guess that would be AlphaBeta search. Why is this no longer computationally feasible exactly? - Note that there still lacks a ... grammar - [Our] AI [system] achieved ... - How many players is the "top 0.04%" ? - l86: shifted to - l87: neural network[s] - l87: defeated [a] top professional ... p3 - General RTS games[:] - l93: in RTS [games] - l96: combine => combined - MOBA games[:] - l112: network[s] like LSTMs Unclear how an LSTM can tackle imperfect information Many more grammar and wording issues - I stopped listing them at this point. The paper requires a thourough proofreading effort to make it publishable in the NeuRIPS proceedings. - develop hierarical action heads this has been done before - reference? p4 - ever-changing game state changing game states are common, why emphasize it? - l174: why introduce multiple value heads when the network should be able to discover such features independently? How important is this incorporated expert knowledge? Would your system work without it? - eq 2: are the w_k fixed or trained? also: why: s.t. \hat V_t = ... ? "s.t." is commonly used as definition or optimization constraint. but \hat V_t isn't p5 - eq 3: why are both losses equally weighted? Are their scales comparable? - l213 ... choosing 5+5 out of 40 using the 1+2+2+2+2+1 picking scheme leads to 192e12 sequences and - and 2.1e11 leaves - The description of the MCTS based drafting algorithm is inadequate. In the current form it can't be reproduced p6 - l240 we develop 9,227 scalar features containing observable unit attributions and in-game stats which are? how? - why did you choose parameters the way you did (e.g. \eps, c ...) Did you do optimize them somehow? - why is the observation delay so long (133 ms) ? p7 - Fig 3. Phase 1 and Phase 2 experiments need to be run for 192/480h to see what a difference CSPL actually makes p8 - l321: what is the ELO value for never-acting bot? - l323: how many games were played between the different picking strategies? References: - use proper capitalization e.g. dota => DOTA monte-carlo => Monte-Carlo bayesian => Bayesian ai => AI


Review 3

Summary and Contributions: This work trains a superhuman agent to play in 5vs5 MOBA game Honor of Kings. It is tested with a pool of 40 heroes, and achieves convincing win rates against top professional players and top amateur players. To achieve this feat, the agent uses off-policy adaption, multi-head value estimation, curriculum self-play learning, policy distillation, and Monte-Carlo tree-search, on top of existing works. Despite the high computation costs, substantial amount of the ablation studies are done, to demonstrate the importance of certain components of the algorithm.

Strengths: It is impressive that authors build a large-scale system to conquer the MOBA game Honor of Kings. Hero pool is increased to 40, leading to much more combinations in game strategies and mechanics, surpassing the 17 hero pool that OpenAI Five use. While OpenAI Five's methods struggles to scale beyond, this work successfully trains a superhuman agent with convincing evaluation results, beating top professional players 40-2 and top amateur players (97.7% win rate in 642k matches, only top 0.04% players allowed to participate). The latter achievement is more impressive in my opinion, demonstrating AI is robust to possible exploitations. Most of the components are not new, but it is novel to use them in real-time MOBA games. The designed curriculum to handle the large hero size pool problem plays a significant role. Although the consumed computing resources are huge, the authors provide enough details on the architecture and parameters, and I believe it can be reproduced given enough resources and engineering efforts. Substantial amount of the ablation studies are done, to demonstrate the importance of certain components of the algorithm.

Weaknesses: One obvious limitation is that the computing resources, which makes this work harder to reproduce. Nevertheless I believe this work can bring significant insights with their detailed description of their system architecture and a qualitative analysis of its playing strength. The other concern is the fairness of the competition - The authors have already listed delays in reaction time and action per minutes distributions. In RTS games sometimes AI agents still have an advantage because all agent APMs are EAPMs (effective actions). This is regularly discussed around StarCraft 2 related works. Do the authors have any stats on this? Also, it seems that the AI agent has full observations on every details of the map including invisible opponent information, does this form an unfair competitive advantage? ------- Post-rebuttal In general I am satisfied with author response on competition fairness and I am confident the authors can fix the grammar and typos.

Correctness: The claim and the methods are sound with strong evaluation results.

Clarity: The paper in general is well written. I suggest the authors to expand a little more details in the core algorithm part especially in 3.3, such as RL playing details in CSPL.

Relation to Prior Work: OpenAI Five is the most similar related work in this setting. The authors clearly discussed what areas are similar as well as the novel contribution of this work.

Reproducibility: Yes

Additional Feedback: nit: CPSL -> CSPL in figure 4. A lot of etc. are used, for example in L132 and L139 it can be illustrated with more details. L130 Scalar feature -> a scalar feature or scalar features, there are multiple occurrences across the paper. L224 Monte-Carole misspell Please proofread one more time to fix the typos etc.


Review 4

Summary and Contributions: This paper presents a complete framework for a 5v5 MOBA game-playing agent, including details for the network architecture and training processes. In addition, it describes a large scale human machine experiment involving both professional and amateur level players, with a total of over 640,000 games for the latter.

Strengths: Subjectively, I would categorize this paper as more of an engineering achievement rather than a strictly scientific one. It is no small feat to train and deploy a full 5v5 MOBA-playing agent (with a 40 playable character roster, which is much more than the previous state-of-the-art). The enormous number of games played against amateur level humans is also impressive, though sadly impossible to replicate at a similar scale for most other research teams. For these reasons I would say the work presented is both significant and highly relevant to the community.

Weaknesses: Following the above comments, the main drawback would also be this work's engineering-oriented nature. The paper is organized in a way that reads like a list of design choices and features, or a collection of previously known techniques/algorithms. For this reason it is not very self-contained, and requires frequent referencing to previous publications. It is also very difficult to distinguish between novel methods proposed in this paper with previous works. For example, in many sections the authors make references to [39] (Ye et al., 2019), which I cannot help but suspect is what this paper is based on. I can understand the difficulty in citing previous works while simultaneously respecting the rules for anonymity, and given the circumstances, I believe there was an adequate effort. However, it would still be immensely helpful if the authors made a stronger effort to compare between the work in [39] and for this paper. For now, my understanding is that the 5v5 curriculum learning part is completely novel, and that the MCTS associated with the so-called "drafting" process is also novel. From this assumption my opinion is that the work contains enough novelty to warrant a new submission, but I would encourage the authors to illustrate this more clearly for a better read.

Correctness: As an engineering report, my opinion is that it is excusable that the proposed agent is almost entirely justified using empirical results. For example, top human player evaluations have been the main measure of success for AlphaGo, AlphaStar, and OpenAI Five. That said, further experiments that analyze the different techniques used in this project are needed to make the paper more scientific. The ablation studies included are a good start, but I would like to know, for example, how generalizable CSPL is for other problem domains, and MOBA is too complex, and involves too many moving parts that interact with each other, so that it is difficult to attribute the difference in performance entirely to CSPL. At the same time, given the limited scope a conference paper can cover, I suspect a better way to present this work would be to focus on specific novel parts of the program for a conference such as this one, with additional experiments to justify these methods' generalizability. Then as a follow up, compile all the moving parts in the project into a full journal paper.

Clarity: As discussed somewhat in the previous parts of my comments, I think the organization of the paper leaves plenty to be desired if the goal is to transform it from an engineering report to a scientific paper. Again, it is understandable that to make such a large endeavor work, many existing techniques need to be incorporated. That said, sections 3.1 and 3.2 read like a laundry list of existing techniques, where the focus should be on novel components instead. The paper is in sore need of native English proofreading. While overall the points are clear enough, there are many grammatical mistakes and some typos.

Relation to Prior Work: As discussed above, I do not think there is enough effort to distinguish the novel parts from previous work in this paper. Regarding section 2 (Related Work), I see what the authors are aiming for by including historical achievements that tackle problems comprehensively such as DeepBlue and AlphaGo. However, the relevance of these projects to the current one is low. I would remove mentions of non-RTS games completely and focus on RTS/MOBA, or even remove RTS and leave just MOBA if the current work is not based on rule-based methods (in RTS) or techniques used in AlphaStar (which I think is indeed the case). That way there is more space for the authors to compare between previous MOBA research with this current work, and can therefore improve the issue regarding separation of old and new methods.

Reproducibility: No

Additional Feedback: Questions: 1) How many distinct players are there for the 640,000+ collected games? For example, winning against the same player 620,000+ times out of 640,000+ times is very different from winning against 640,000+ opponents. I imagine the numbers are large enough to answer any doubts to whether the win rates are representative of agent strength, but it would help to have this statistic in the paper. 2) Is the MCTS drafting match dataset created after RL training for the program has concluded? Or is it collected while the program is training? How diverse is this dataset in terms of matchups? Are there any plans to analyze the interaction between drafting and gameplay itself? A small sample of grammatical errors, typos, and minor comments: - line 205: distils -> distills - line 224: Monte-Carole -> Monte-Carlo - line 9: methodologically (do you mean methodically?) - line 86: attentions have been -> attention has been (attention is not countable) - line 158: so as to simply the correlations between action heads (I don't know how to fix this, perhaps "so as to simplify?) - lines 175 to 178: what is no-op? what are tyrant buff, dark tyrant buff, etc.? My suggestion would be to keep this vague and general, or to cover the details in the appendix. - line 278: counter a diversity of high-level strategies -> counter diverse high-level strategies - line 309: results in unacceptable slow training -> results in unacceptably slow training

[Author Response · NeurIPS 2020]

We thank all the reviewers for the constructive comments. We address the major concerns below.

**Q1.** Reproducibility: 1) learning to draft details; 2) feature details; 3) discussions on the computing resources used.

**1)** When drafting, a search tree is built iteratively, with each node representing a state and each edge representing the
action (pick a hero) resulting to a next state. The search tree is updated based on four steps of MCTS. The selection,
expansion and backup steps are the same as the normal MCTS. For the prediction step, we use a value network to predict
the value of the current state. After 1600 iterations, we pick heroes based on the child of the root node with the most
visits. For the win-rate predictor and value network, the dimensions of 3 FC layers are [200,128],[128*2,128],[128,1]
and [200,128],[128*2+1,128],[128,1], respectively. The learning rate is set to 0.001 with Adam. The MCTS drafting
match dataset contains about 30 million matchups, covering all the heroes in our hero pool uniformly. The source code
for drafting will be released.  **2)** We have added more details on the feature design, including the range of features,
feature dimension, the way of normalization, and how these features were developed.  **3)** We believe that using a large
amount of training resource is not a weakness of the proposed method, considering the nature of the problem we solve.
First, the computing resources required for complex game-playing AI programs are often non-trivial, e.g., AlphaGo
(280 GPUs), OpenAI Five (1920 GPUs), AlphaStar (3072 TPUv3 cores), and this work (320 GPUs). Second, we will
publish our infrastructure (in progress). Third, we will develop subtasks of MOBA-game-playing for the AI community.

**Q2.** Novelty: the building blocks have been reported in the literature, e.g., curriculum learning, MCTS, etc.

The way we use the building blocks and the problem we solve are both new, which has been clearly pointed out
by R1, R3 and R4. This work's focus is on an unsolved MOBA-game-playing problem, rather than the standalone
building blocks themselves. Individually, each of the employed techniques has been innovatively adapted. For example,
curriculum learning is just a conceptual framework; how to design a curriculum for MOBA AI remains unexplored.

**Q3.** Fairness concerns: 1) AI's reaction compared to human player; 2) use of full information in the value network.

**1)** MOBA is different from general RTS. In RTS like StarCraft, a player controls many game units, and one can always
choose to switch to other game units whose skills and attacks are usable. In this case, it is a must to restrict the EPMs.
By comparison, in MOBA, one player controls one hero. All the hero skills have cooldown limit and the normal attack
has attack-frequency-limit. In this case, reaction-time is generally used as the measurement for fairness in MOBA.
OpenAI Five has a discussion on this (see Section 4.1 in their paper). "In Dota 2, the key measure of human dexterity is
reaction time, contrast with RTS games like StarCraft..."  **2)** Please be assured that this will not cause any unfairness.
In our model, we ONLY use the full information in the value network. As we mentioned in Section 3.2–Value Updates:
"Note that this is performed only during training, as we only use the policy network during evaluation."

**Q4.** Presentation & organization: 1) the paper is organized in a way that reads like a list of design choices and features;
2) more comparisons to a recent MOBA AI work by Ye et al. [39]; 3) removing the non-RTS part in related work.

**1)** As suggested by R4, we will improve the presentation in two ways: 1) further emphasizing the newly proposed parts,
e.g., CSPL, MHV, and drafting; 2) re-organizing the sections in Method in an easier-to-read fashion.  **2)** This work and
[39] have completely different focuses. [39] studies MOBA 1v1 game. while this paper studies the scalability-related
issues when expanding the hero pool in MOBA 5v5 game. The similarities between this work and [39] include: the
modeling of action heads (the value heads are different); off-policy correction; the way of conducting human evaluation.
We will make it clearer for a better read.  **3)** We will remove the non-RTS part in related work.

**Q5.** CSPL related: 1) the rule of advancing to next curriculum; 2) the non-CSPL experiments haven't been run long
enough to make a clear call that they won't achieve similar ELO values; 3) in Fig 3., Phase 1 and Phase 2 experiments
need to be run for 192/480h to see what a difference CSPL actually makes.

**1)** The rule of advancing to the next curriculum in CSPL is based on the convergence of Elo scores. The convergence in
student-driven policy distillation is not sensitive to the hero pool size. Both 20-hero and 40-hero cases can converge
rapidly within two days.  **2)** First, Fig. 3-d aims to show that CSPL can achieve faster learning than the non-CSPL
counterpart (we can already observe CSPL speeds it up). Second, we kept running it for another 3 days; still, the
non-CSPL did not achieve a similar Elo score.  **3)** Phase 1 and Phase 2 are parts of the CSPL method itself.

**Q6.** Various minor issues: 1) Computation time in petaflop/s; 2) #heroes in OpenAI Five; 3) Minimax or AlphaBeta
search, why computationally infeasible; 4) How many players in top 0.04%; 5) Why introduce multiple value heads; 6)
Why the observation delays so long. 7) Elo of the never-acting bot; 8) #games played between the different picking
strategies? 9) Unclear how LSTM can tackle imperfect information. 10) Grammar, phrasing and reference problems.

**1)** 1.523 PFlops/s·days.  **2)** 17.  **3)** Minimax is used in OpenAI Five; AlphaBeta pruning is not vanilla Minimax;
because of the 213,610,453,056 combinations in a 40-hero case.  **4)** 120,000.  **5)** For better value estimation, as stated.
**6)** To make the AI's reaction speed comparable to human players.  **7)** 0.  **8)** 1000.  **9)** we meant to say that it can help
infer unobservable information.  **10)** We have done a thorough proofreading.

[Meta-Review · NeurIPS 2020]

This paper demonstrates an application of RL and search to a challenging MOBA game-playing task, leading to AI agents able to defeat top professional human players. Three out of four reviewers consider that although this is an application-oriented paper with a strong engineering focus, it is still relevant enough for publication at NeurIPS. Only R2 is advocating for rejection, based essentially on the lack of scientific novelty. I believe that such impressive large scale applications of RL are well worth pushing forward and I am thus recommending acceptance. The general algorithms being used may not be novel, but their instantiation to solve this specific task largely is. In addition, although the exact techniques developed here may be specific to MOBA games, I expect them to potentially inspire further applications both within and outside of video games (I do not believe, like R2 suggested, that there is a need for a second application in this submission, since at such scale each application domain is worth a paper on its own IMHO). The authors’ feedback helped clarify many points raised by the reviewers, and the authors committed to improve the paper accordingly (please do so! This will be very helpful to future readers of your paper)